# Influence of Psychological Factors on Vaccination Acceptance among Health Care Workers in Slovenia in Three Different Phases of the COVID-19 Pandemic

**DOI:** 10.3390/vaccines10121983

**Published:** 2022-11-22

**Authors:** Vislava Globevnik Velikonja, Ivan Verdenik, Karmen Erjavec, Nevenka Kregar Velikonja

**Affiliations:** 1Division for Obstetrics and Gynaecology, University Medical Centre Ljubljana, Šlajmerjeva 4, 1000 Ljubljana, Slovenia; 2Faculty of Health Sciences, University of Novo Mesto, 8000 Novo Mesto, Slovenia

**Keywords:** healthcare workers, COVID-19, vaccination intention, vaccination, anxiety, perceived infectability

## Abstract

COVID-19 vaccination acceptance among healthcare workers (HCWs) is very important to control the pandemic and to ensure the safety of HCWs and patients. As psychological factors may affect the decision to be vaccinated, the aim of this study was to investigate the influence of psychological factors on vaccination acceptance in different phases of the COVID-19 pandemic. A cross-sectional study using a web-based survey was conducted among HCWs in Slovenia at the beginning of the pandemic (N = 851), one month later (N = 86), and one year later (N = 145) when vaccines were already available. The results showed that the influence of psychological factors (anxiety, psychological burden, perceived infectability, and germ aversion) was specific for each survey period. At the beginning of the pandemic, vaccination intention was positively associated with anxiety. In the third survey period, anxiety was not exposed as a predictive factor for vaccination intention. However, comparison of vaccination status among groups with different levels of anxiety revealed an interesting distinction within those in favour of vaccination; in the group with minimal levels of anxiety, there was a relatively high share of respondents that were already vaccinated, whereas in the group with severe anxiety, most individuals intended to be vaccinated but hesitated to take action.

## 1. Introduction

Globally, the threat of the coronavirus pandemic has caused great changes in people’s lives. People face concerns about loss of income, fear of infection and death, social isolation and lockdown, which affects their behavior, and concerns about physical and mental health. The daily worries associated with COVID-19 are relevant to the person’s psychological maladjustment, their ability to cope with the stress associated with various potentially dangerous conditions, and with the likelihood of developing depression and/or anxiety symptoms [1,2,3,4,5]. In the general population, older adults are the least worried; they expressed lower fears [5].

Healthcare workers (HCWs) are at a higher risk than any other group due to constant contact with patients and infections, a lack of personal protective equipment, and inadequate infection control training [6,7]. They are more likely to contract the disease than members of the general public [8] and they are at a high risk of transmitting the disease [9]. Scientists and practitioners agree that the acceptance of the vaccine against COVID-19 by HCWs is very important to control the pandemic and to ensure the safety of HCWs and patients. However, HCWs attitudes towards COVID-19 vaccines vary worldwide, with vaccination acceptance ranging from 4.3% to 77.3% [10,11,12,13]. Most of the studies have found that the individuals who were males, of older age, doctoral degree holders, and those who had been vaccinated against influenza before had significantly higher intentions in being vaccinated against COVID-19 [10,11,12,13]. The results of the meta-analysis of cross-sectional studies of COVID-19 vaccination acceptance among HCWs published in 2020 and 2021 show moderate acceptance and no significant relationship between occupation and intention to be vaccinated against COVID-19 [14]. However, some studies have reported that medical staff being less qualified may be associated with the refusal to take up COVID-19 vaccination, e.g., physicians are more likely to accept vaccination compared to nurses [12,15,16].

HCWs facing critical situations, such as the COVID-19 pandemic, who are aware of a potential infection are exposed to considerable stress, and this intensely experienced stress brings about psychological problems [6,7]. In terms of the psychological impact of COVID-19, HCWs are vulnerable to infection risks, increased workloads and transmission to their families [17]. Staff in contact with patients had higher levels of both acute or post-traumatic stress (PTSD) and psychological distress [18]. In addition, in the general population, people who sensed a greater risk of COVID-19 infection were more likely to have adaptive behaviors such as washing their hands and maintaining social distancing [19]. Risk interpretation is influenced by cognition, affective reactions, and contextual factors; therefore, the relationship between societal adaptation and vaccine attitudes has been complicated. In one study, societal adaptation was found to be directly related to vaccine worries and indirectly related to vaccine worries by the mediation of PTSD [17]. Current studies have revealed that HCWs experienced increased anxiety and fear and showed increased depressive symptoms during the COVID-19 pandemic [20,21,22,23,24,25,26]. There are few studies comparing HCWs’ mental health between the first and second phase of COVID-19 [26,27,28,29,30], with contradictory results when it came to higher levels of anxiety, stress, and depression.

A systematic review and meta-analysis of the factors related to the intention of HCWs to accept COVID-19 vaccination shows that fear of COVID-19, an individual’s perceived risk related to COVID-19, and lower self-satisfaction related to COVID-19 were related to higher COVID-19 vaccination acceptance [31]. A German study on the influence of psychological factors on the adult population’s intention to be vaccinated shows that COVID-19-related fears, fears of infection, and health consequences were significantly positively correlated with vaccine acceptance, whereas the broader constructs of non-specific fears and depressive symptoms were not significantly associated with vaccine acceptance [32]. In contrast, a Japanese longitudinal study among the adult population examined whether generalized trust, mental health factors such as depression and generalized anxiety, and fear of COVID-19 were predictors of vaccine hesitancy against COVID-19 in autumn 2020 and spring 2021: (1) respondents who had lower levels of generalized trust in both research periods were more likely to be undecided or unwilling to be vaccinated in spring 2021; (2) respondents with moderately severe or severe depression in both research periods were more likely to be undecided about being vaccinated in spring 2021; (3) respondents who had moderate or severe generalized anxiety in spring 2021 but not in autumn 2020 were more likely to be undecided or unwilling to be vaccinated in spring 2021, and (4) participants who had high levels of fear of COVID-19 in both research periods were less likely to be undecided or unwilling to be vaccinated in spring 2021, when vaccines were already available [33].

There are only a few studies on the influence of psychological factors on the vaccination intention of HCWs. A study from the Kingdom of Saudi Arabia showed that HCWs with lower generalized anxiety disorder (GAD-7) scores were more likely to accept the new vaccine in the first phase [34]. A Japanese study reported that depressive symptoms were more pronounced in HCWs who were unwilling to be vaccinated against COVID-19 than in those who were willing to be vaccinated [35].

The COVID-19 outbreak affected different countries over a three-year period and varied in intensity depending on the availability of vaccines and other factors. While there are many studies on vaccination intention among HCWs, some of these studies were conducted before COVID-19 vaccines became available, whereas others were conducted after the vaccine became available. Therefore, it is still unclear how the different phases of the COVID-19 pandemic will affect HCWs’ vaccination intention. This raises the question of how vaccination intentions among HCWs have changed and what psychological factors influence their decisions.

In Slovenia, the pandemic was declared from 12 March 2020, to 1 June 2020, and again from 18 November 2020, to 16 June 2021. The massive vaccination strategy started on 27 December 2020, and HCWs were on the priority list as one of the most vulnerable groups. During the last period of our survey (March–May 2021), the percentage of the vaccinated population raised from 6.9% to 30.6% [36]. During this period, there were many public debates regarding vaccine safety, also due to the temporary suspension of some vaccines based on reports of extremely rare side effects, and the importance of active surveillance to evaluate the short-term and long-term side effects and effectiveness of various COVID-19 vaccines was emphasised [37].

Although comprehensive models of various factors have been developed to test HCWs’ vaccination intention, psychological factors that may have influenced the vaccination process remain unexplained. For the vaccination programme to be successful, the reasons for HCWs’ vaccination hesitancy need to be identified. Thus, it is necessary to also investigate the influence of psychological factors on the vaccination intention and vaccination of HCWs in different phases of the COVID-19 pandemic. The aim of this study was therefore to investigate longitudinal changes in (a) vaccination intention and vaccination, (b) psychological factors (anxiety, psychological burden, perceived infectability, and germ aversion) and (c) the influence of psychological factors, education, age, and living arrangements on vaccination intention at different phases of the pandemic, namely at the beginning and during the first phase of the pandemic, when there was only awareness of the possible benefits of vaccination to limit the spread of the pandemic in society, and one year later, when vaccines were already available and the strategy of mass vaccination was fully implemented by the state.

## 2. Materials and Methods

### 2.1. Study Design

A cross-sectional study with a web-based survey on COVID-19 vaccination intention, and attitudes towards vaccination was conducted among HCWs in Slovenia at the beginning of the pandemic (1st survey period), one month later (2nd survey period), and one year later, when vaccination was already available (3rd survey period).

The online survey was prepared using the web-based platform 1 ka [33], and it was distributed among HCWs via professional contacts and further distributed using the snowball sampling method [7,8,9,10,11,12,13,14,15,16,17,18,19,20,21,22,23,24,25,26,27,28,29,30,31,32,33,34,35,36,37,38,39]. Participants were asked to share the survey link with their colleagues who were willing to take part in the study. Respondents were asked to complete a self-administered, structured electronic questionnaire.

The study protocol was reviewed and approved by the Ethical Committee at the Faculty of Health Sciences, University of Novo Mesto (University ethical approval No. FZV-98/2020). Data were collected with the voluntary participation of anonymous participants.

### 2.2. Participants

Only the participants who identified themselves as HCWs were selected for analysis. In the first survey period, the questionnaire was completely filled out by 7764 respondents; 14% of the respondents (N = 851) belonged to the healthcare sector. In the second survey period, the response rate was much lower; of the 313 respondents, 86 were HCWs. In the third survey period, the questionnaire distribution was strictly limited to the HCW population; 145 responses were appropriate for further analysis. The demographic characteristics of the respondents are presented in Table 1. There were no significant differences among the respondent groups in gender, age, and education characteristics. However, there were some differences in living arrangements among the groups.

### 2.3. Research Instrument

The questionnaire included 21 questions containing 134 variables on the demographic characteristics of the participants (age, gender, level of education, work engagement, if they have already had COVID-19), adherence to preventive behaviour, vaccination intention, vaccination acceptance, and attitudes, as well as the psychological burden, anxiety, germ aversion, and perceived infectability. It took 10 to 14 min for the respondents to complete the survey.

The parts of the research instrument were as follows:Respondents’ vaccination intention and vaccination status: in the first and second survey period, the respondents were only asked if they would be vaccinated if the vaccine was available (response options: yes/I do not know/no). In the third survey period, the respondents were asked whether they were already vaccinated (response options yes/no) and about their intention to be vaccinated (response options yes/not sure/no).Psychological burden: the feeling thermometer (FT) was used to assess the psychological burden, i.e., one’s own experience of physical, emotional, psychosocial burden, and the burden of everyday life during the last 7 days. Respondents had to assess these on a continuous visual scale from 0 (no burden) to 10 (extremely strong burden). The FT provides a summary score that investigators use under the expected utility and information theory, and it has shown good properties for the measurement of health-related quality of life (HRQL). The results suggest moderate reliability of clinical marker states’ ratings for the FT [40,41].Anxiety: The generalized anxiety disorder 7-item, GAD-7 [35], was used. The GAD-7 consists of seven questions based in part on the DSM-IV (Diagnostic and Statistical Manual of Mental Disorders–Fourth Edition) criteria for GAD and reflects the frequency of symptoms during the preceding 2-week period; for each symptom queried, it provides the following response options: “not at all”, “several days”, “over half the days”, and “nearly every day”, and these are scored 0, 1, 2, or 3, respectively (Cronbach’s alpha = 0.92). The GAD-7 scores range from 0 to 21, with higher scores indicating progressively higher levels of anxiety (0–4 = minimal anxiety, 5–9 = mild anxiety, 10–14 = moderate anxiety, and 15–21 = severe anxiety). When used as a screening tool, further evaluation is recommended when the score is 10 or greater. Using the threshold score of 10, the GAD-7 has a sensitivity of 89% and a specificity of 82% [42,43].Perceived infectability and germ aversion: The perceived vulnerability to disease questionnaire, PVDQ, developed by Duncan et al. [44], was adapted to be more reflective of the current reality [45], and a 15-item self-report on a 7-point scale response (with endpoints labeled as “strongly disagree” and “strongly agree”) was used. It measures two factors: perceived infectability (assesses beliefs in one’s own susceptibility to infectious diseases, e.g., “If an illness is going around, I will get it”; seven items) (Cronbach’s alpha = 0.87), and germ aversion (assesses emotional discomfort in contexts where disease-causing germs might be transmitted, e.g., ”It really bothers me when people sneeze without covering their mouth”; eight items) (Cronbach’s alpha = 0.74) [38]. In our study, Cronbach’s alpha showed lower internal consistency of these scores (0.68 and 0.56, respectively).

### 2.4. Statistical Analysis

The data obtained were coded, validated, and analyzed using SPSS (IBM SPSS Statistics for Windows, Version 27, IBM Corp., Armonk, NY, USA). Descriptive analysis was used to calculate frequencies and proportions. To assess an association between the variables, Spearman correlation was performed. A T-test, Mann–Whitney test, ANOVA, and Kruskal–Wallis test were used to assess the differences between the groups made according to different demographic characteristics. In multiple comparisons in post-hoc ANOVA, a Bonferroni correction was applied.

For the Likert scales, we used parametric versions of the tests, whereas for two ordinal categorical questions (vaccination intention and advising vaccination), we chose non-parametric versions (Kruskal–Wallis for multiple groups, Mann–Whitney for two groups).

A *p*-value less than 0.05 was considered statistically significant.

## 3. Results

During the survey periods, the percentage of respondents who declared vaccination hesitancy (either explicitly ‘I do not intend to be vaccinated’ or ‘I am not sure’) decreased. In the third period, 31% of the respondents were already vaccinated, whereas an additional 42.1% of the respondents declared their intention to be vaccinated (Table 2). For the purpose of further analysis and for comparison of the factors among the survey period, which were influencing the pro-vaccination attitude, these two groups were merged (Table 3 and Table 4). Additionally, the respondents that were already vaccinated and those that intended to be vaccinated but had not taken action yet were compared to investigate the factors influencing the decision to be vaccinated in the third survey period (Table 5 and Table 6).

### 3.1. Analysis of Factors Influencing Vaccination Intention in Different Survey Periods

The measured psychological factors were changing in different periods of the survey, as shown in Table 3. The highest perceived infectability among HCWs was measured in the third survey period, whereas germ aversion, GAD-7, and psychological burden decreased. There were no statistically significant differences in anxiety between the survey periods.

Table 4 and Figure 1 show the results of the regression analysis of the influence of psychological factors, education, age, and living arrangements on the intention to be vaccinated in different phases of the pandemic. At the beginning of the pandemic, the intention to be vaccinated was affected by anxiety and education level. Those with a higher level of anxiety expressed a greater intention to be vaccinated. The same was observed in HCWs with a postgraduate education. In the third survey period, vaccination intention was influenced by perceived infectability, education level, and age. HCWs who expressed a higher level of threat due to possible infection, those with a higher education level (postgraduate education), the elderly (middle adulthood (30–44), and those in late adulthood (45 years and older) expressed a greater intention to be vaccinated or had already been vaccinated. Living arrangements did not have a statistically significant influence on the intention to be vaccinated.

### 3.2. Comparison of Respondents Who Were Already Vaccinated and Respondents Who Expressed Their Intention to Be Vaccinated in the Third Survey Period

Differences in the measured psychological factors were observed between the groups of respondents who were already vaccinated and those who declared their intention to be vaccinated but had not taken action yet (Table 5). In all of the factors, the values were higher in the group that was not yet vaccinated, and a statistically significant difference was found in the factors of perceived infectability and anxiety.

Table 6 and Figure 2 show the results of the regression analysis of the influence of psychological factors, education level, living arrangements, and age in the third survey period on the decision to be vaccinated among those who expressed their intention to be vaccinated: the results illustrate the differences between the group of respondents who were already vaccinated and the group of respondents who declared their intention to be vaccinated (but had yet to be vaccinated) in the third survey period. Perceived infectability and age influenced the decision to be vaccinated. Older respondents who were already vaccinated had a significantly lower degree of fear of infection.

### 3.3. Level of Anxiety and Vaccination Intention

Anxiety did not differ significantly between the observed periods; in the first survey period it only appeared as a factor influencing vaccination intention in the regression analysis. Therefore, the associations of vaccination intention in relation to the levels of anxiety were further investigated.

Table 7 shows that there was a statistically significant difference in the intention to be vaccinated among groups of respondents with different levels of anxiety in the first and third survey period. In the first period, respondents with higher levels of anxiety expressed their intention to be vaccinated more often. In the third period, the relationship between the level of anxiety and vaccination acceptance was more complex. In the group with minimal levels of anxiety, there was a relatively low share of respondents that did not intend to be vaccinated; this share was significantly higher in the groups with moderate and severe anxiety. However, in respondents with severe anxiety, most individuals intended to be vaccinated but hesitated to take action.

Table 8 shows that the correlation between anxiety and other psychological scores (psychological burden, perceived infectability, and germ aversion) was different according to vaccination intentions in all three survey periods. In all three periods, there was a significantly high correlation between anxiety and psychological burden regardless of vaccination intention. The relationship between anxiety and perceived infectability was different from one survey period to another. In the first period, the correlation between these scores was high regardless of the intention to be vaccinated, in the second this correlation was no longer significant, and in the third, a significantly negative correlation between anxiety and perceived infectability appeared among those who did not intend to be vaccinated. The relationship between anxiety and germ aversion was less pronounced: in the first period, there was a weak significant correlation between anxiety and germ aversion in the respondents who expressed their intention to be vaccinated, and in the third period, this weak correlation was observed in those who did not intend to be vaccinated.

## 4. Discussion

This study aimed to find the answer to the central research question of what influence psychological factors and their longitudinal changes had on the vaccination intention of HCWs and the levels of vaccination among HCWs at the beginning and in the early phase of the pandemic when society was just becoming aware of the potential benefits of vaccination to control the pandemic, and one year later when vaccines were already available and the strategy of massive vaccination had been fully implemented by the state. The results suggest that the influence of psychological factors (anxiety, psychological burden, perceived infectability, and germ aversion) was specific in each survey period. This means that the influence of psychological factors on the changes in HCWs’ vaccination intentions within each observed period of the COVID-19 pandemic was determined by specific health and social situations and other personal characteristics of the HCWs, as the attitudes towards vaccination had specific cognitive, emotional, and conative components.

Using an integrated behavioral model, Indonesian researchers found out that health care workers’ intention to obtain COVID-19 vaccines was associated with favorable vaccine attitudes, perceived norms, and self-efficacy. Among the determining constructs, behavior belief predicted vaccination intention the best [45].

Before elaborating on the impact of psychological factors on HCWs’ vaccination intentions, the changes in the psychological factors in each survey period should be explained. The perceived infectability of HCWs differed statistically significantly according to the phase of the pandemic. The feeling of being threatened by a possible infection was greater when the pandemic was declared than one month later. This can be explained by the fact that when the COVID-19 pandemic was declared (the first phase), HCWs had little knowledge of the disease, protective equipment was lacking, and the health system had to quickly introduce new rules for working in health facilities and new clinical pathways for treating the infected and the sick [46,47]. After one month, the sense of threat among HCWs was slightly lower, as the state provided protective equipment for HCWs and the population, and the recommendations for testing the population and treating the infected and the sick in separate rooms were already in place [47]. Perceived infectability was statistically significantly higher in the third survey period than in the second survey period. The feeling of being exposed to possible infection was highest among HCWs in the third survey period (one year after the start of the COVID-19 pandemic) when the predominant perception was that we were all at risk, when everybody knew someone with COVID-19, and when many HCWs had already contracted COVID-19 [48], although vaccination was already available [49]. According to our results, 38.7% of HCWs had already contracted COVID-19 and 1.9% had symptoms but did not know whether they had COVID-19 in the third survey period.

Germ aversion also differed statistically significantly between the phases of the pandemic, with it being highest at the beginning of the pandemic and lowest after one year, which can be explained by the fact that fear of the unknown was prevalent at the beginning, as it was not yet clear how the virus was transmitted, the media advised complete isolation, the quarantine of purchased items for several days was advised, etc. [7]. One year later, much more was known about how the virus was transmitted, and disinfection of hands, work surfaces, handles, and tools became routine within the health system, and HCWs were also taught how to protect themselves from infection [48]. Germ aversion was statistically significantly lower in the third survey period than in the first survey period.

Psychological burden, i.e., one’s own experience of physical, emotional, and psychosocial burden, and the stress of everyday life in the last 7 days decreased statistically significantly among HCWs during the pandemic. While the number of cases and deaths escalated, the pandemic also significantly disrupted and changed our daily lives, especially with the extreme measures taken to prevent the spread of the disease in different regions. HCWs reported the greatest psychological distress at the beginning of the pandemic, when the country went into lockdown, classes were held remotely, children were cared for at home, many people worked from home, and HCWs were instructed to be present at work. Similarly, research from India comparing two phases of the pandemic reported that HCWs were less affected by psychological impacts during the second phase [49]. A statistically significantly lower psychological burden in the third survey period than in the first survey period could be explained by psychological resilience. The risk that stress and negative life events trigger mental illness has long been known. People react differently to any change, but most of them generally adapt well to such stressful events over time. These positive responses or outcomes in the face of significant risk or adversity are generally known as resilience [50]. Resilience has been recognised as a protective factor in reducing stress in HCWs during the COVID-19 pandemic as well [51].

There were no statistically significant differences in anxiety among the survey periods phases. Similarly, two Italian studies found no differences in anxiety among HCWs between the first and the second phase of the pandemic [42,51], which was expressed as generalized anxiety in 13% to 15% of HCWs [52]. In our study, the percentage of anxiety (moderate and severe) among HCWs ranged from 14.9% in the first survey period, 17.2% in the second phase, and 11.0% in the third phase (14.6% for all survey periods). The measured prevalence of anxiety in HCWs during the COVID-19 pandemic varies significantly among different studies, e.g., from 8.7% in a multicentre study in Singapore and India [53] to 37% in a meta–analysis of 44 published studies [54], which may be partly the consequence of different criteria, measuring instruments, workplaces during the pandemic, and intercultural differences. Studies on the global prevalence of anxiety disorders in the general population showed a prevalence increase of 25.6% in the first year of the pandemic [55].

As expected, the intention of HCWs to be vaccinated increased over time. At the beginning of the pandemic, 38.6% of HCWs expressed their intention to be vaccinated, and this rose to 43.7% one month later and 73.1% after one year. By the time of the third survey period, the vaccine was already available, and 31.7% of HCWs had already been vaccinated, which is slightly higher than the average vaccination rate in the general population, which increased from 6.9% to 30.6% at the time of the third survey period [36].

The results also show that at the beginning of the pandemic, the intention to be vaccinated was influenced by anxiety and education level. Those who were more anxious expressed a stronger intention to be vaccinated, as did HCWs with a higher education level (postgraduate education). In the second survey period, the intention to be vaccinated was significantly influenced by postgraduate education, and in the third period, it was significantly influenced by perceived infectability, education level, and age. The HCWs who expressed a higher degree of threat due to possible infection, those with a higher education level (postgraduate education), and the elderly (middle adulthood, 30–44 years old and late adulthood, 45 years and older) expressed a greater intention to be vaccinated or were already vaccinated. It seems that people over 45 years were aware of their greater risk, but those over 30 were also responsible, as by then most of them already had a family or took care of their elderly parents. This is consistent with the findings of a recent Italian study showing that fear of COVID-19, its consequences, and the likelihood of isolation are greater in young adults compared to older people, which could be explained by lower levels of anxiety, depression, and stress and greater preventive measures in older people [5]. Education level has a significant influence on the intention to be vaccinated, which has also been confirmed by other studies performed in different phases of the pandemic [46,47,48,49,50,51,52,53].

The structure of the respondents according to their living arrangements differs among the survey periods; however, our findings show that the living arrangements have no statistically significant effect on HCWs’ intention to be vaccinated. Although living arrangements have been proposed as an important factor among the various factors influencing vaccination intention [56], HCWs are exposed to a high infection risk in their working environment and the vaccination decision is influenced by several other important factors, such as self-protection, protection of patients, and work environment recommendations.

The results of the regression analysis of the influence of psychological factors, education level, living arrangements, and age in the third survey period on actual vaccination among those who expressed their intention to be vaccinated show that the decision to be vaccinated was influenced by perceived infectability and age. Older HCWs more often decided to be vaccinated, and those who were vaccinated perceived the risk of infection to be significantly lower. The results also showed that those who had already been vaccinated expressed lower levels of anxiety and lower levels of perceived infectability compared to those HCWs who intended to be vaccinated but had not yet been vaccinated. The positive impact of vaccination against COVID-19 on mental health and anxiety levels has also been confirmed by other studies [57,58].

On the other hand, those HCWs who delayed vaccination, despite having had the opportunity to be among the first people to be vaccinated as one of the more vulnerable populations, reported higher levels of anxiety, which may also be due to the information about the side effects of the vaccine, uncertainty about the safety of the vaccine, and conspiracy theories. Some studies show evidence of links between distrust in the vaccine and vaccine hesitancy and resistance [59].

The question is what the cause is and what the effect is—whether anxious HCWs are reluctant to be vaccinated because they are afraid of being vaccinated, or whether they are more fearful of being unvaccinated because they are not protected. Due to the complexity of the relationship between anxiety and vaccination, we decided to analyze the decision to vaccinate (‘I do not intend to be vaccinated’, ‘I intend to be vaccinated’, and (relevant only in third period) ‘I have already been vaccinated’) according to the level of anxiety.

Although anxiety did not differ significantly among the observed periods, it appeared as a factor influencing vaccination intention only in the first survey period; more anxious HCWs expressed higher vaccination intention at the beginning of the pandemic. However, interesting associations between vaccination intention and anxiety levels were found in the third survey period when vaccines were already available.

In the third survey period, the majority of HCWs (66.2%) had minimal levels of anxiety symptoms, 22.8% reported mild anxiety, 6.9% reported moderate anxiety, and 4.8% reported severe anxiety. In the vaccinated group, there was a very high percentage of HCWs with minimal levels of anxiety (82.2%). In the groups with mild and moderate- anxiety, the percentage of vaccination opponents was relatively high (45.5% and 40.0%, respectively) compared to those with minimal and severe anxiety (20.8% and 0.0%, respectively). Among those who intended to be vaccinated, the higher the level of anxiety, the longer the uptake of vaccination was delayed despite the expressed intention. The group with severe symptoms was small but very homogenous in expressing their vaccination intention, yet they did not decide to be vaccinated.

Given that the least anxious individuals are the most likely to accept vaccination as one of the most effective protective measures, this indicates a greater capacity for adaptive behaviour in more emotionally stable individuals, which makes decisions regarding preventive measures easier and faster. At all stages of the research, we confirmed the expected significant positive correlation between anxiety and psychological burden regardless of vaccination intention. Two-thirds of HCWs remained psychologically stable even in the third survey period, without significant symptoms of anxiety and psychological distress, which indicates a high level of resilience to stress in the majority of HCWs. Greater attention should be directed to those who express greater psychological distress, as their anxiety is also greater.

We have found that the relationship between anxiety and perceived infectability changes with the observed time periods. In the first survey period, the correlation between them was high regardless of vaccination intention, while in the third survey period, a negative correlation between anxiety and perceived infectability appeared among those who did not intend to be vaccinated. The greater the level of anxiety in this group, the lower their perceived risk of infection. This might be explained by the fact that the individuals who refused vaccination and advocated natural immunity even tried to purposely be infected in order to avoid vaccination and regular testing, which was obligatory at the time of the survey. Interestingly, in the same group, a weak significant correlation between anxiety and germ aversion was observed. It is possible that in the same group there were heterogeneous factors influencing these scores; while some wanted to acquire natural immunity, others were afraid of infection for other reasons (e.g., those with chronic diseases, weaker immune systems, the elderly, and those who should not be vaccinated).

Risk interpretation was influenced by cognition (education level and information), affective reactions (anxiety, psychological burden, perceived infectability, and germ aversion), and contextual factors (different phases of the pandemic, age, and individual family arrangement). Therefore, the relationship between adaptive behaviors and vaccine attitudes was complex.

In the study, 26.9% of the HCWs did not intend to be vaccinated one year after the COVID-19 outbreak, which can be a major challenge as HCWs play an important role in the attitudes of the general population towards vaccination. A similar study revealed that 22.5% of HCWs had hesitations about the COVID-19 vaccine [11]. Compared to the general population, this is a somewhat lower percentage: in a systematic review of 16 studies with 30,242 participants from the general population, the prevalence of COVID-19 vaccine hesitancy was 33.2% (95% CI 24.7–41.4%) [60].

Future studies and social policies should direct more attention to the group of unvaccinated HCWs, with additional health-related education and psychological support in cases of recognized higher psychological burden and anxiety. Considering the fact that we have to accept the changing epidemiological virus situation, as well as other global crises (climate change, wars) that require long-term changes in attitudes and habits, it makes sense to focus especially on the younger generation, who show a greater willingness to change their behavior and habits compared to the older population, by educating them and providing them with the right information [5].

In the fight against infodemia (an abundance of true and false information that overwhelms the subject by depriving him or her of the ability to process information appropriately) and misinformation, as well as the rapid spread of fake news through various social media platforms, we should take advantage of these media to provide the public with evidence-based information, even though this role has so far been played by traditional media sources. At the same time, public health actors (ministries of health, public health institutes, and the Centers for Disease Control and Prevention) should ensure that people are not just informed but also receive guidance on how to act appropriately, as was also proposed by Zaracostas [61].

A major limitation of this study is that the sample of HCWs in the different periods of the survey was only roughly comparable, as we could not obtain the same sample due to the conditions of the COVID-19 pandemic (e.g., illness, overtime, and job changes). The online survey used snowball sampling, which does not allow for control over respondent selection and estimation of the response rate. The other limitation of this study is that psychological outcomes were determined using a self-report tool.

## 5. Conclusions

Mental stability is a key factor that enables resilience to stress, adaptive behaviors, and fact-based decision-making especially in crisis situations such as the COVID-19 pandemic. The study shows that determinants of vaccine acceptance changed over the course of the pandemic. Despite the fact that during the duration of the pandemic confidence in vaccines (intention to be vaccinated) gradually increased, 27% of HCWs refused to be vaccinated when the vaccine was already available. Attention should be directed to the group of unvaccinated HCWs, with additional health-related education and psychological support in cases of recognized higher psychological burden and anxiety.

HCWs play an important role in educating the general public about COVID-19 and vaccination and in dispelling the existing conspiracy theories. Since the findings show that the influence of psychological factors is specific in each phase of the COVID-19 pandemic, multiple strategies are needed to maintain the psychological stability of HCWs, which support their resilience to stress and their adoption of appropriate adaptive behaviors.

## Figures and Tables

**Figure 1 vaccines-10-01983-f001:**
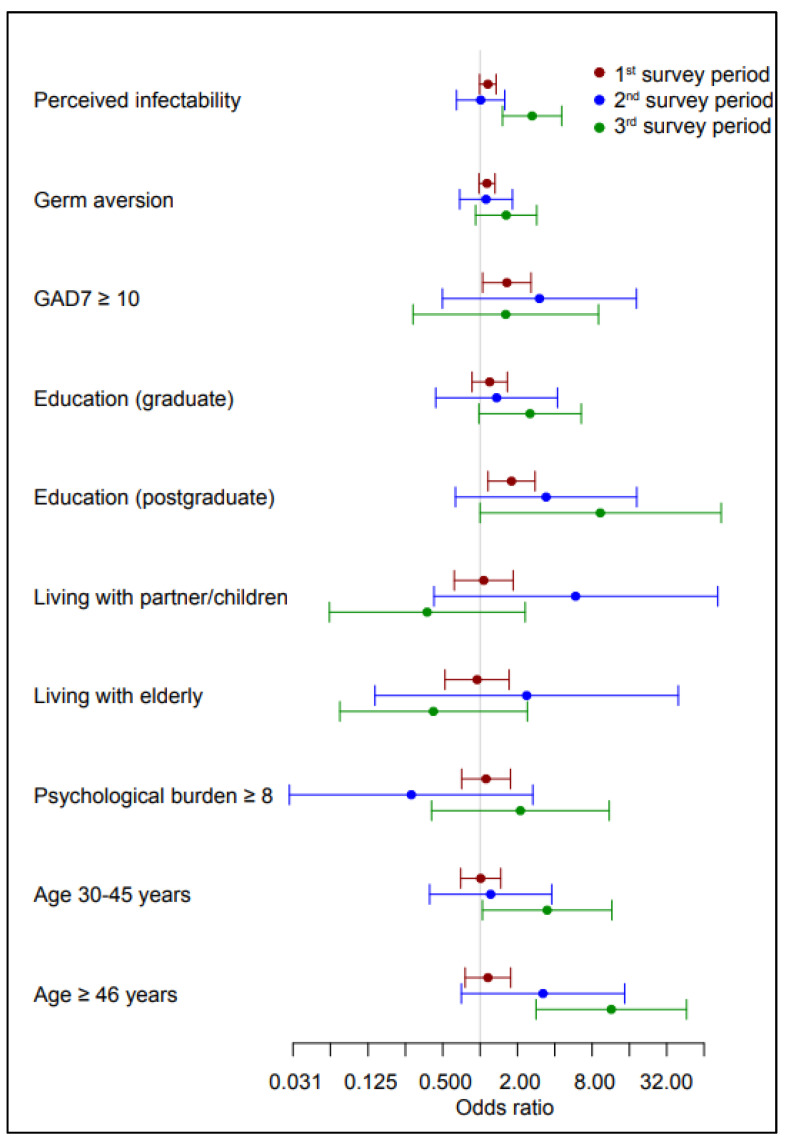
Regression analysis of factors influencing vaccination intention in the survey periods. Odds ratios above 1 denote greater intention to be vaccinated.

**Figure 2 vaccines-10-01983-f002:**
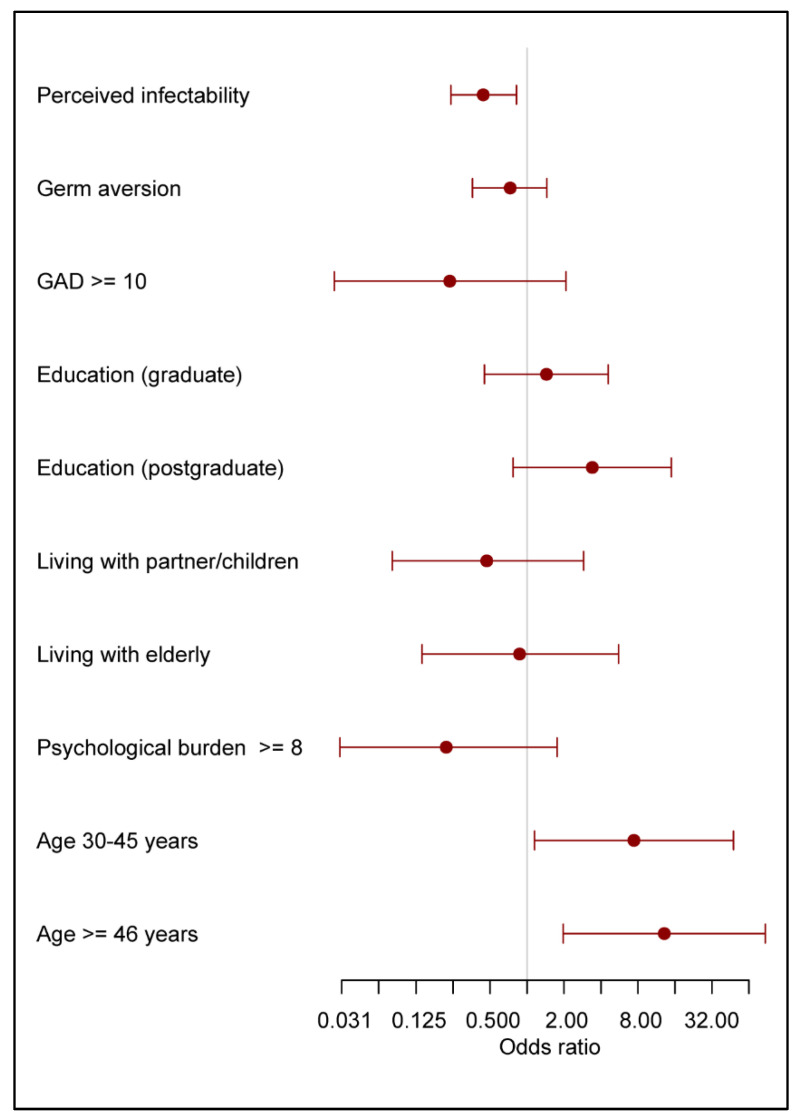
Regression analysis of factors influencing decision for vaccination and vaccination intention in the third survey period. Odds ratios above 1 denote greater proportion of already vaccinated to those still hesitating.

**Table 1 vaccines-10-01983-t001:** Time frame of survey activity and demographic characteristics of respondents.

Survey Period		I	II	III	Difference between Samples: Chi-Square (*p*)
Period		13 March 2020–14 March 2020	13 April 2020–8 May 2020	7 March 2021–26 May 2021	
No. of respondents (N)		7764	313	154	
No. of health care workers		851	86	145	
Gender N (%)	Female	742 (87.2%)	75 (87%)	125 (86.8%)	0.947
Education N (%)	Secondary school	357 (41.7%)	46 (52.9%)	67 (46.2%)	0.149
	Graduate	362 (42.3%)	26 (29.9%)	60 (41.4%)	
	Postgraduate	137 (16.0%)	15 (17.2%)	18 (12.4%)	
Age N (%)	>29 years	198 (23.7%)	33 (38.8%)	37 (25.5%)	0.051
	30–45 years	420 (50.4%)	35 (41.2%)	70 (48.3%)	
	>46 years	216 (25.9%)	17 (20.0%)	38 (26.2%)	
	Alone	71 (8.3%)	6 (6.9%)	15 (10.3%)	0.006 *
Living arrangements (%)	Living with partner/children	570 (66.5%)	60 (69.0%)	74 (51.0%)	
	Living with elderly	216 (25.2%)	21 (24.4%)	56 (38.6%)	

* *p* < 0.05.

**Table 2 vaccines-10-01983-t002:** Vaccination intention and vaccination status of the respondents in different survey periods.

Survey Period	Vaccination Intention Not Declared N (%)	Vaccination Intention Declared N (%)	Already Vaccinated N (%)	Total N
I	526 (61.4%)	331 (38.6%)	-	857
II	49 (56.3%)	38 (43.7%)	-	87
III	39 (26.9%)	61 (42.1%)	45 (31.0%)	145

**Table 3 vaccines-10-01983-t003:** Comparison of the psychological scores in different survey periods (ANOVA and Bonferroni post-hoc test).

	Period I	Period II	Period III	*p* (ANOVA)	Bonferroni Post-Hoc Test Significance: Periods (*p*)
Number of cases	857	87	145		
Perceived infectability	3.45 ± 1.02	3.22 ± 1.11	3.61 ± 0.89	0.015 *	II-III (0.012)
Germ aversion	4.97 ± 1.01	4.89 ± 1.04	4.69 ± 0.76	0.007 *	I-III (0.005)
GAD-7	4.59 ± 5.16	4.62 ± 5.36	3.89 ± 4.57	0.296	
Psychological burden	4.66 ± 2.51	4.13 ± 2.74	4.04 ± 2.65	0.009 *	I-III (0.021)

* The mean difference is significant at the 0.05 level.

**Table 4 vaccines-10-01983-t004:** Regression analysis of factors influencing vaccination intention in different survey periods.

Phase	Factor	B	Sig. (*p*)	Odds Ratio (95%CI)
I	Perceived infectability	0.144	0.065	1.16 (0.99–1.35)
	Germ aversion	0.128	0.092	1.14 (0.98–1.32)
	GAD 7 (≥10)	0.497	0.030	1.64 (1.05–2.58)
	Education (secondary school)		0.032	1 (ref)
	Education (graduate)	0.178	0.286	1.20 (0.86–1.66)
	Education (postgraduate)	0.584	0.009	1.79 (1.16–2.78)
	Living alone		0.783	1 (ref)
	Living with partner/children	0.067	0.810	1.07 (0.62–1.85)
	Living with elderly	−0.056	0.854	0.95 (0.52–1.72)
	Psychological burden (≥8)	0.111	0.632	1.12 (0.71–1.76)
	Age ≤ 29 years		0.737	1 (ref)
	Age 30–45 years	0.012	0.950	1.01 (0.70–1.47)
	Age ≥46 years	0.145	0.501	1.16 (0.76–1.77)
II	Perceived infectability	0.011	0.962	1.01 (0.65–1.58)
	Germ aversion	0.111	0.657	1.12 (0.69–1.82)
	GAD 7 (≥10)	1.103	0.230	3.01 (0.50–18.25)
	Education (secondary)		0.359	1 (ref)
	Education (graduate)	0.307	0.595	1.36 (0.44–4.21)
	Education (postgraduate)	1.226	0.153	3.41 (0.63–18.35)
	Living alone		0.217	1 (ref)
	Living with partner/children	1.773	0.186	5.89 (0.43–81.70)
	Living with elderly	0.865	0.547	2.38 (0.14–39.64)
	Psychological burden (≥8)	−1.276	0.268	0.28 (0.03–2.66)
	Age ≤29 years		0.302	1 (ref)
	Age 30–45 years	0.196	0.734	1.22 (0.39–3.78)
	Age ≥46 years	1.168	0.131	3.22 (0.71–14.65)
III	Perceived infectability	0.966	0.001	2.63 (1.52–4.55)
	Germ aversion	0.484	0.094	1.62 (0.92–2.86)
	GAD 7 (≥10)	0.476	0.588	1.61 (0.29–9.01)
	Education (secondary school)		0.046	1 (ref)
	Education (graduate)	0.928	0.056	2.53 (0.98–6.54)
	Education (postgraduate)	2.232	0.050	9.32 (1.00–87.03)
	Living alone		0.560	1 (ref)
	Living with partner/children	−0.983	0.290	0.37 (0.06–2.31)
	Living with elderly	−0.866	0.330	0.42 (0.07–2.40)
	Psychological burden (≥8)	0.749	0.373	2.11 (0.41–11.00)
	Age ≤29 years		0.003	1 (ref)
	Age 30–45 years	1.246	0.042	3.48 (1.05–11.55)

**Table 5 vaccines-10-01983-t005:** Comparison of psychological scores between HCWs who were already vaccinated and those who only expressed an intention to be vaccinated (***T***-test).

	Vaccinated	Intended to Be Vaccinated	ANOVA (*p*)
Number of cases	45	61	
Perceived infectability	3.42 ± 0.93	4.00 ± 0.83	0.001 *
Germ aversion	4.72 ± 0.82	4.76 ± 0.62	0.751
GAD-7	2.47 ± 3.32	4.57 ± 5.64	0.028 *
Psychological burden	3.47 ± 2.73	4.26 ± 2.70	0.139

* The mean difference is significant at the 0.05 level.

**Table 6 vaccines-10-01983-t006:** Regression analysis of factors influencing decision for vaccination in third survey period.

Survey Period	Factor	B	Sig. (*p*)	Odds Ratio (95%CI)
III	Perceived infectability	−0.824	0.010	0.44 (0.24–0.82)
	Germ aversion	−0.320	0.366	0.73 (0.36–1.45)
	GAD7 ≥ 10	−1.449	0.193	0.23 (0.07–2.08)
	Education (secondary school)		0.267	1 (ref)
	Education (graduate)	0.362	0.541	1.44 (0.45–4.58)
	Education (postgraduate)	1.223	0.106	3.40 (0.77–14.96)
	Living alone		0.470	1 (ref)
	Living with partner/children	−0.767	0.411	0.47 (0.08–2.89)
	Living with elderly	−0.135	0.887	0.87 (0.14–5.56)
	Psychological burden ≥ 8	−1.523	0.153	0.22 (0.03–1.76)
	Age ≤ 29 years		0.029	1 (ref)
	Age 30–45 years	2.004	0.035	7.42 (1.15–47.94)
	Age ≥ 46 years	2.573	0.008	13.11 (1.97–87.10)

**Table 7 vaccines-10-01983-t007:** Vaccination status and intention in relation to the level of anxiety in the three survey periods.

Survey Period	Level of Anxiety	N	Vaccination Intention Not Declared (%)	Vaccination Intention Declared (%)	Already Vaccinated (%)	Chi-Square (*p*)
I	Minimal (0–4)	537	65.9%	34.1%		0.001 *
	Mild (5–9)	192	57.8%	42.2%		
	Moderate (10–14)	64	50.0%	50.0%		
	Severe (15–21)	64	45.3%	54.7%		
	Total	857	61.4%	38.6%		
II	Minimal (0–4)	53	58.5%	41.5%		0.838
	Mild (5–9)	19	47.4%	52.6%		
	Moderate (10–14)	8	62.5%	37.5%		
	Severe (15–21)	7	57.1%	42.9%		
	Total	87	56.3%	43.7%		
III	Minimal (0–4)	96	20.8%	40.6%	38.5%	0.006 *
	Mild (5–9)	33	45.5%	33.3%	21.2%	
	Moderate (10–14)	10	40.0%	60.0%	0.0%	
	Severe (15–21)	6	0.0%	83.3%	16.7%	
	Total	145	26.9%	42.1%	31.0%	

* The difference between groups is significant at the 0.05 level.

**Table 8 vaccines-10-01983-t008:** Correlation between anxiety and other psychological scores (psychological burden, perceived infectability, and germ aversion) according to vaccination status and intention in all three survey periods.

Survey Period			Pearson Correlation Coefficient between Anxiety and Other Psychological Scores:		
	Group of Respondents	N	Psychological Burden	Perceived Infectability	Germ Aversion
I	Vaccination intention not declared	526	0.539 **	0.374 **	0.072
	Vaccination intention declared	331	0.545 **	0.236 **	0.133 *
II	Vaccination intention not declared	49	0.679 **	0.004	0.097
	Vaccination intention declared	38	0.600 **	0.254	0.108
III	Vaccination intention not declared	39	0.419 **	−0.508 **	0.333 *
	Vaccination intention declared	61	0.479 **	0.163	−0.089
	Already vaccinated	43	0.590 **	−0.131	−0.282

** The correlation is significant at the 0.01 level (2-tailed). * The correlation is significant at the 0.05 level (2-tailed).

## Data Availability

The data presented in this study are available on request from the corresponding author.

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
