# Peer review of "Influence of Psychological Factors on Vaccination Acceptance among Health Care Workers in Slovenia in Three Different Phases of the COVID-19 Pandemic"

_vaccines, 2022, doi:10.3390/vaccines10121983_

Round 1

Reviewer 1 Report

Thank you for giving me the opportunity to review the manuscript with title: " Influence of psychological factors on vaccination acceptance among health care workers in different phases of the COVID-19 pandemic" by Velikonja V.G. et al. 

This study has very interesting results, giving us the possibility to understand reasons for lack of vaccination against COVID-19 among HCWs. The manuscript is accepted for publications after minor corrections.

The authors are describing that " In the second survey period, the intention to be vaccinated was significantly influenced by postgraduate education", lines 217-218, however on table 4, regression analysis results, p-value is not <0.05. It is proposed to check numbers or rephrase.

Lines 382-385 could be in the previous paragraph, where the authors describe the effect of education to the intention of vaccination.

All the references should have the same format, according to Journal instruction.

Author Response

Dear reviewer,

We appreciate your precious time in reviewing our paper and providing valuable comments. It was your valuable and insightful comments that led to possible improvements in the current version. The authors have carefully considered the comments and tried our best to address every one of them.

We hope the manuscript after careful revisions meet your high standards. The authors welcome further constructive comments if any. Below we provide the point-by-point responses.

All modifications in the manuscript have been highlighted with track-changes. Text was edited by English professional.

Authors

 Comment 1: The authors are describing that "In the second survey period, the intention to be vaccinated was significantly influenced by postgraduate education", lines 217-218, however on table 4, regression analysis results, p-value is not <0.05. It is proposed to check numbers or rephrase.

Response: 

We thank reviewer for noticing this. Indeed, the presented result was not relevant, and we deleted it from the text (p.6, lines 233-234).

Comment 2: Lines 382-385 could be in the previous paragraph, where the authors describe the effect of education to the intention of vaccination.

Response: 

We agree with the reviewer's suggestion. The discussion was restructured in line with this suggestion and verification of results in previous comment (p. 13, lines 395-397).

Comment 3: All the references should have the same format, according to Journal instruction.

Response: 

The references were checked again and corrected according to Journal instruction.

Reviewer 2 Report

First of all, I am grateful for the opportunity to review this paper.

WHAT IS THE MAIN QUESTION ADDRESSED BY THE RESEARCH? Currently, the world is facing the threat of the coronavirus pandemic. Actually, the vaccination campaign is the first method to counteract the COVID-19 pandemic; however, sufficient vaccination coverage is conditioned by the people’s acceptance of these vaccines, especially in some groups of population such as HCW. Moreover, daily worries, depression and anxiety may affect vaccination acceptance. In this context, the authors aimed to investigate the influence of psychological factors in different phases of the COVID-19 pandemic on vaccination acceptance.

IS THE TOPIC ORIGINAL OR RELEVANT IN THE FIELD? The subject under study is certainly important, especially in the historical period we are experiencing. The article presents interesting results but, it must be improved, especially for some methodological concerns.

WHAT SPECIFIC IMPROVEMENTS SHOULD THE AUTHORS CONSIDER REGARDING THE METHODOLOGY?

Title: It can must be improved reporting the time and place where the study was conducted.

Introduction: the authors must consider the confounding factors related to psychological maladjustments a part from COVID-19 that may have affected the sample, the paper must be first of all included in the frame of the current emerging priorities raising anxiety, depression etc. (refer to articles with DOI: 10.3390/ijerph191911929). Then they must report the knowledge already existing and that they will consider. Finally, they must better explain their aims.

Methods:

1. The authors do not talk about a minimum sample size, and it is not clear what is the reference population: all Slovenian HCW? How large is the reference population? Without the numerical identification of the reference population is not clear the validity of the study. A non-representative sample is by its self a non-sense-survey.

2. The selection of the sample raises many criticisms. It is not completely clear how was it selected. How did this method allowed to avoid selection bias. This still requires detailed explanation.

3. The authors compare the results of three different periods of the pandemic, but the selected people seem different. Therefore, no comparison can be significant, also due to the incomparable samples of the three surveys.

4. The survey was conducted in part (point 1 and two) using a non-standard questionnaire. The use of an unreliable instrument is a serious and irreversible limitation of the study. The fact that the questionnaire construction refers to previously used surveys is not sufficient. A validation process must be performed to evaluate the tool. What about reliability, intelligibility and validation index?

ARE THE CONCLUSIONS CONSISTENT WITH THE EVIDENCE AND ARGUMENTS PRESENTED AND DO THEY ADDRESS THE MAIN QUESTION POSED? I also suggest expanding. The discussion must be enriched considering the confounding factors related to psychological maladjustments, a part from COVID-19, that may have affected the sample (see the above-mentioned reference). What are the future prospects originating from this paper? especially, in terms of Public Health. Please examine in depth these aspects.

Author Response

Dear reviewer,

We appreciate your precious time in reviewing our paper and providing valuable comments. It was your valuable and insightful comments that led to possible improvements in the current version. The authors have carefully considered the comments and tried our best to address every one of them.

We hope the manuscript after careful revisions meet your high standards. The authors welcome further constructive comments if any. In the attached document we provide the point-by-point responses.

All modifications in the manuscript have been highlighted with track-changes. Text was edited by English professional.

Authors

Round 2

Reviewer 2 Report

The paper was improved and it is now suitable for publication